# Fully Dynamic Embedding into $\ell_p$ Spaces

**Kiarash Banihashem** [1]  **Xiang Chen** [2]  **MohammadTaghi Hajiaghayi** [1]  **Sungchul Kim** [2]  **Kanak Mahadik** [2]
**Ryan A. Rossi** [2]  **Tong Yu** [2]

## Abstract

Metric embeddings are fundamental in machine learning, enabling similarity search, dimensionality reduction, and representation learning. They underpin modern architectures like transformers and large language models, facilitating scalable training and improved generalization. Theoretically, the classic problem in embedding design is mapping arbitrary metrics into $\ell_p$ spaces while approximately preserving pairwise distances. We study this problem in a fully dynamic setting, where the underlying metric is a graph metric subject to edge insertions and deletions. Our goal is to maintain an efficient embedding after each update. We present the first fully dynamic algorithm for this problem, achieving $O(\log(n))^{2q} O(\log(nW))^{q-1}$ expected distortion with $O(m^{1/q+o(1)})$ update time and $O(q \log(n) \log(nW))$ query time, where $q \geq 2$ is an integer parameter.

## 1. Introduction

Metric embeddings are fundamental to computer science, with applications in machine learning, databases, and network design. In ML, they are critical for tasks like similarity search, dimensionality reduction, and representation learning, underpinning modern architectures such as transformers and large language models (Vaswani, 2017; Kenton & Toutanova, 2019). By enabling efficient and context-aware representations of high-dimensional data, embeddings drive scalable training and task generalization, making them essential for advancing state-of-the-art models.

From a theoretical perspective, the classic formulation of metric embedding problem is as follows. We are given a set of points in an input metric space $M = (V, d)$, and we want to map (i.e., *embed*) each point $v \in V$ to a point $v' \in V'$ in a target metric space $M' = (V', d')$. While this can be

done in many ways, the usual goal in embedding design is to ensure small *distortion*, i.e., we make sure that the distances are approximately preserved.

Low-distortion embeddings with "simple" target metric spaces $V'$ effectively simplify the original metric while approximately preserving distances. This can be useful from an algorithm design perspective as quite often, the problem is much more approachable in the simpler metric, especially from a computational perspective. Because of this, many works have previously studied such embeddings for a variety of target metrics such as $\ell_p$ spaces (Bourgain, 1985) and hierarchical trees (Bartal, 1996a; 1998; Fakcharoenphol et al., 2004). A closely related line of work also examines the problem of maintaining a low-stretch spanning tree of a graph (Alon et al., 1995; Elkin et al., 2008). These embeddings have been used to obtain algorithms for a variety of problems such as sparsest cut (Linial et al., 1995), network design (Awerbuch & Azar, 1997; Garg et al., 2000), minimum bandwidth (Blum et al., 1998), and metric labeling (Kleinberg & Tardos, 2002)

Embeddings are a cornerstone of modern machine learning, powering models from word2vec to large language models (LLMs). These methods represent discrete data—such as words, entities, or tokens—in continuous metric spaces, enabling geometric reasoning about semantic relationships. This makes embeddings useful for tasks such as node classification, link prediction, and community detection (Zhang & Chen, 2018; Abu-El-Haija et al., 2018; Yun et al., 2021; Davison et al., 2024) Among embedding targets, $\ell_p$ spaces (especially $\ell_2$) have emerged as the standard representation space in much of modern machine learning. This is due to (a) their interpretability—distances and angles have clear geometric meaning—and (b) the fact that many key computational primitives are substantially more efficient in $\ell_p$ spaces. For example, nearest neighbor search in high dimensions is routinely performed using locality-sensitive hashing (LSH), a technique specifically designed for $\ell_2$ and other vector norms.

In the past decade, motivated by the surge of interest in modern Big Data applications, there has been a renewed interest in studying classical problems in a *dynamic* setting, where the underlying input is constantly changing and the goal is to

---

[1]University of Maryland, College Park [2]Adobe research. Correspondence to: Kiarash Banihashem <kiarash@umd.edu>.

*Proceedings of the $42^{nd}$ International Conference on Machine Learning*, Vancouver, Canada. PMLR 267, 2025. Copyright 2025 by the author(s).

always maintain a good solution. This is considerably more difficult than the *static* setting, where the entire input is fixed and given upfront, as the algorithm needs to quickly adapt to (adversarial) changes in the data. For dynamic maintenance of low-stretch spanning trees, (Forster & Goranci, 2019) obtained an algorithm for unweighted graphs with $n^{1/2+o(1)}$ update time and $n^{o(1)}$ average stretch. The update time was further improved to $n^{o(1)}$ by (Chechik & Zhang, 2020). This was achieved by developing a new pruning technique for obtaining Low Diameter Decompositions (LDDs), which partition the graph into clusters with bounded diameter such that the number of inter-cluster edges is low. Building on this pruning technique, (Forster et al., 2021) obtained a dynamic algorithm for embeddings into probabilistic trees which has $O(\log(n))^{2q}O(\log(nW))^{q-1}$ expected distortion and $O(m^{1/q+o(1)})$ update time where $q \geq 2$ is an integer parameter. A separate line of work studies online algorithms for embeddings, where the focus is generally on making irrevocable decisions rather than minimizing computation (Indyk et al., 2010; Barta et al., 2020; Newman & Rabinovich, 2023).

In this work, we focus on the problem of dynamically embedding into $\ell_p$ spaces. In the static setting, the seminal result of (Bourgain, 1985) shows that any metric can be embedded into $\ell_p$ space with $O(\log n)$ distortion and using $O(\log^2 n)$ dimensions. This was later shown to be tight by (Linial et al., 1995). Recently, (Banihashem et al., 2024) studied this problem in a restricted dynamic setting, referred to as the *decremental* setting, which only allows for the edge weights to be increased. They obtained an algorithm $O((m^{1/2+o(1)}\log^2(W)+Q\log(n))\log(nW))$ total update time and $O(\log^3 n)$ expected distortion, where $W$ refers to the largest weight in the graph and $Q$ denotes the number of updates. Their techniques do not generalize to the *fully dynamic* setting, where distances can both increase and decrease; the main idea behind their approach is to directly turn the information provided by the decremental LDDs of (Forster et al., 2021) into coordinates for the $\ell_p$ space and it is not clear whether a fully dynamic version of these LDDs can be obtained. They additionally obtain a lower bound showing that any algorithm that maintains an embedding that has low distortion with high probability, and explicitly outputs all changes the embedding of each vertex, cannot be fully dynamic.

In this paper, we circumvent this negative result by providing a fully dynamic algorithm for the problem which maintains the changes *implicitly* via a function $\rho$ that can efficiently be queried, and has low *expected* distortion. Our embedding has the additional guarantee that it is non-contractive with high probability. We further strengthen the lower bound of (Banihashem et al., 2024) by extending to embeddings with low expected distortion that are non-

contractive with high probability, showing that an efficient algorithm with these properties cannot be obtained in the fully dynamic model.

## 1.1. Our Results

We study a fully dynamic setting in which the edges of a graph $G = (V, E)$ are inserted and deleted, and each edge has some weight in the range $\{1, \ldots, W\}$.[1] Since we study low-distortion embeddings into the $\ell_p$ metric where the distance between any two points is never $+\infty$, we will assume that any two vertices in different connected components are in distance $W' \gg nW$ (whereas normally one would assume that their distance is $+\infty$). Intuitively, this is assuming that there is always a "default" edge between any two vertices with a very high weight, even if there is no edge $e \in E$ connecting them. [2]

Our goal is to randomly maintain a low-distortion embedding function $\rho : V \to \mathbb{R}^\ell$ that can efficiently calculate the embedding of a queried vertex. We note that, while our choice of the embedding function is random, the function $\rho$ itself is a deterministic mapping. In other words, while changing the random seed of the algorithm will change the function $\rho$ but for any fixed seed, the function $\rho$ will not change unless the graph changes.

Our main result is the following theorem.

**Theorem 1.1.** *Let $G = (V, E)$ be a weighted undirected graph undergoing edge insertions and deletions. For any $q \geq 2$, there exists an algorithm $A$ (see Algorithm 1) that randomly maintains an embedding function $\rho : V \to \mathbb{R}^\ell$ with the following properties:*

- **Low expected distortion and non-contractivity.** *For any two vertices $u, v$,*

$$1 \leq \frac{\mathbb{E}\left[|\rho(u) - \rho(v)|\right]}{d_G(u,v)} \leq O(\log(n))^{2q}O(\log(nW))^{q-1},$$
(1)

*where the expectation is over the randomness in choosing $\rho$. Furthermore, with high probability[3], the embed-*

---

[1] In this work, we consider the *oblivious adversary* model, in which the updates are specified by an adversary who knows our algorithms but does not have access to the random bits we use; see Section 2 for more details.

[2] The infinite distances is also noted by (Banihashem et al., 2024), who study the decremental case. They handle this issue by assuming that the input graph is originally connected and only allowing edge weight increases (and not deletions) to ensure the graph remains connected.

[3] Throughout the paper, we use the term with high probability to denote probabilities $\geq 1 - 1/n^a$ where $a$ is an arbitrarily large constant.

*ding is non-contractive; i.e.,*

$$\|\rho(u) - \rho(v)\| \geq d_G(u, v) \text{ for all } u, v \in V \quad (2)$$

- **Efficient update time.** *the amortized update time of the algorithm is $m^{1/q+o(1)}$ with high probability.*

- **Efficient query time.** *For any vertex $v \in V$, we can calculate $\rho(v)$ in time $O(q \log(nW) \log(n))$.*

We further show that if we require that each update to the embedding is explicitly outputted, then no such result can be obtained, even if the expected distortion is only required to be sublinear.

**Theorem 1.2.** *Assume that the algorithm $A$ is a fully dynamic algorithm for maintaining a random embedding $\rho : V \to \mathbb{R}^k$ with the following guarantees:*

1. *$**Sublinear expected distortion and non-contractivity*$: $d_G(u, v) \leq \mathbb{E}[\rho(u) - \rho(v)] \leq o(W')d_G(u, v)$ and the embedding is non-contractive with probability at least $3/4$.*

2. *$**Explicit changes*$: any change to the embedding of a vertex $v$ is explicitly outputted by the algorithm.*

*The amortized update time of the algorithm $A$ is at least $\Omega(n)$.*

### 1.2. Overview of Techniques

We next present an overview of our proofs. We start by briefly recapping the standard approach for constructing $\ell_p$ embeddings in the static setting. We then examine a natural algorithm based on this approach and explain why it fails in the dynamic setting. Finally, we present a new algorithm based on these failures and briefly outline its proof of correctness.

**Recap of static approach.** We start by briefly recapping the standard approach for embedding into $\ell_p$ spaces. Set $h = \lceil \log n \rceil$. For each $i \in [h]$, we obtain the set $S_i$ by uniformly sampling the vertices in $V$ with probability $p_i := \frac{1}{2^i}$. For any vertex $v$, we define its embedding to be the concatenation of its distances from $S_i$.[4] Formally, $\rho(v) := (d_G(S_1, v), \ldots, d_G(S_h, v))$, where $d_G(S, v)$ denotes the distance between $v$ and the closest vertex in $S$; i.e., $d_G(S, v) := \min_{v_S \in S} d_G(v_S, v)$. We claim that $\rho$ satisfies the desired bounds for the expected distortion. We focus on the $\ell_1$ norm for simplicity; the results for the other norms follow directly at the cost of an additional log factor given the inequality $\|\rho(u) - \rho(v)\|_p \leq \|\rho(u) - \rho(v)\|_1 \leq \log(n)\|\rho(u) - \rho(v)\|_p$.

---

[4] The standard construction uses $\log n$ independent copies for each $S_i$ to ensure high probability bounds; for simplicity we skip this step as our main focus is on expectation bounds.

The upper bound follows from the triangle inequality for the metric $d_G$. Fix any two vertices $u$ and $v$. For any $i \in [h]$ we have

$$|d_G(S_i, u) - d_G(S_i, v)| \leq |d_G(u, v)| \quad (3)$$

by the triangle inequality; summing over all $i$ we obtain $\|\rho(u) - \rho(v)\|_p \leq \|\rho(u) - \rho(v)\|_1 \leq h|d_G(u, v)|$.

The proof of the lower bound is more involved. The main idea is to show that the sets $S_i$ are likely to be "close" to one of $u$ and $v$ and "far" from the other. Specifically, let $r_i$ to be the smallest radius such that $|B(u, r_i)|, |B(v, r_i)| \geq 2^i$ where $B(x, r)$ denotes the (closed) ball with radius $r$ with center $x$. If $r_i \geq d_G(u, v)/2$, then reduce it to $d_G(u, v)/2$; i.e., set $r_i$ to the minimum of the two mentioned values. We note that $r_h = d_G(u, v)/2$ because the minimum radius to include all the points is at least $d_G(u, v)$, and $r_0 = 0$ because $x \in B(x, 0)$ for any vertex $x$. We claim that $\mathbb{E}[|d_G(S_i, u) - d_G(S_i, v)|] \geq c(r_i - r_{i-1})$ for some constant $c$. We note that this implies the lower bound because

$$\mathbb{E}[\|\rho(u) - \rho(v)\|_1] \geq \sum_{i=1}^{h} |d_G(S_i, u) - d_G(S_i, v)|$$

$$\geq \sum_{i=1}^{h} c(r_i - r_{i-1}) = cr_h,$$

which is at least $\Omega(d_G(u, v))$. To prove the claim, we assume that $r_i > r_{i-1}$ since otherwise it holds trivially. By definition of $r_i$, we have $\min(|B(u, r_i - 1)|, |B(v, r_i - 1)|) < 2^i$; assume w.l.o.g that $|B(u, r_i - 1)| < 2^i$. Since $r_i + r_{i-1} - 1 < 2r_i - 1 < d_G(u, v)$, the sets $B_1 := B(v, r_{i-1})$ and $B_2 := B(u, r_i - 1)$ do not intersect. Furthermore, $|B_1| \geq 2^{i-1}$ and $|B_2| \leq 2^i$. Therefore, with constant probability, the set $S$ intersects $B_1$ but does not intersect $B_2$; the probability of each of the two events are constants given the size of the sets and the events are independent because $B_1 \cap B_2 = \emptyset$. When this happens, we have $d_G(S_i, u) \geq r_i$ and $d_G(S_i, v) \leq r_{i-1}$, finishing the proof.

**Candidate dynamic algorithm.** To implement this algorithm dynamically, a natural approach is to use dynamic distance oracles, which can calculate the distance between any two vertices, to obtain $d_G(S_i, u)$. Note that while $S_i$ is a set of vertices and not a single vertex, we can contract the set $S_i$ into a single vertex and run the distance oracle on the contracted graph. The problem with this approach however is that dynamic distance oracles only obtain *approximate* distances and the approximation breaks the triangle inequality in Equation (3). Specifically, denoting the distance estimates with $\hat{d}(.,.)$, if $u$ and $v$ are close to each other but are both far from $S_i$, then a $(1 + \epsilon)$-approximate distance oracle

has an error up to $\epsilon d_G(S_i, u)$ when calculating $d_G(S_i, u)$ and $d_G(S_i, v)$, which means $|\hat{d}_G(S_i, u) - \hat{d}_G(S_i, v)|$ can be much larger than $d_G(u, v)$. The proof for the lower bounds also breaks down; the approximation error for the distance oracle changes the bound on $|d_G(S_i, u) - d_G(S_i, v)|$ to $r_i - (1 + \epsilon)r_i$ which is not desirable and may even be negative.

**Low-distortion trees.** Our solution to the above issues is to obtain the distance estimates by calculating distances in another metric; specifically, distance in trees.[5] This allows us to circumvent the issue with Equation (3) by invoking the triangle inequality in the new metric. Specifically, let $s$ be some fixed vertex in the tree (e.g., the root), and consider the embedding $\rho(u) := d_T(s, u)$. Then for any two vertices $u$ and $v$,

$$|\rho(u) - \rho(v)| = |d_T(s, u) - d_T(s, v)| \leq d_T(u, v),$$

which is small in expectation provided the tree $T$ itself has low expected distortion. This fixes the issue with the proof for the upper bound.

For the lower bound proof, our key insight is to, somewhat counter-intuitively, *inject noise* into the distance estimates. Specifically, instead of working with the tree $T$ directly, we modify it by multiplying the weight for each edge by a random number taken uniformly from $\{1, 2\}$. The main idea here is that if $u$ and $v$ are far from each other, then their path to the root must be very different. Since the multiplication factor is chosen independently for each edge, this means that the noises themselves ensure that the lower bound holds.

Formally, as we show in Section 4, one can break the path $P_{u,v}$ between $u$ and $v$ in the tree $T$ into two parts $P'_u$ and $P'_v$ such that

$$d_{T'}(s, u) - d_{T'}(s, v) = \sum_{e \in P'_u} w_{T'}(e) - \sum_{e \in P'_v} w_{T'}(e),$$

where $T'$ refers to the new tree obtained after randomly modifying edge weights. We note that our choice of $P'_u$ and $P'_v$ does not depend on the scaling factors for $T'$. If the independent noises for the scaling of each edge simply combined, they would equate to $\sum_{e \in P_u} w_T(e) \geq d_T(u, v)$, which would give the desired lower bound. This is not the case in general however because of concentration; the variance in the above difference equals $\sum_{e \in P_{u,v}} \text{Var}(w_{T'}(e))$ which is $\Theta\left(\sum_{e \in P_{u,v}} w_T^2(e)\right)$ given the choice of scaling factors. This can be much less than the desired variance

---

[5]Technically, since the graph $G$ can be disconnected, we will be considering *forests*. Since any forest is simply a collection of separate trees however, we use the terms interchangeably unless the distinction is important (in which case we point this out).

of $d_T^2(u, v) = (\sum_{e \in P_{u,v}} w_T)^2$ which we require for the proofs. To solve this issue, we will rely on a property of existing dynamic tree embeddings which, to the best of our knowledge, was not previously noted. Specifically, we show that the path between any two vertices in $V$ contains a "heavy" edge whose weight is within a small factor of $d_T(u, v)$ itself. This allows us to show that the (undesired) concentration effects do not hold here and we can still recover the lower bound.

**Lower bound.** The proof for the lower bound uses a "bridge graph" which was also used by (Banihashem et al., 2024). Specifically, we consider a graph with two separate cliques of size $n/2$, which we repeatedly connect and disconnect. The fact that the embedding is non-contractive with high probability means that each time the edge is removed, the two sides need to be in distance $W'$. When the edge is added back however, the two sides need to be close again. Using the expected distortion guarantee and Markov's inequality, we conclude that for any fixed $u$ and $v$ that are in different components, with high probability, $\|\rho(u) - \rho(v)\|$ must change after each update, which can only happen if at least one of $\rho(u)$ and $\rho(v)$ change. Since there are $\Omega(n^2)$ pairs, this means that we require at least $\Omega(n)$ changes.

## 2. Preliminaries

**Notation.** For any positive integer $k$, we use $[k]$ to denote the set $\{1, \ldots, k\}$. We use $G = (V, E)$ to denote the weighted graph being updated and use $\rho$ to denote the embedding we maintain. We use $n$ and $m$ to denote the number of vertices and edges in $G$ respectively and the weights are assumed to be integers in the range $[1, W]$. For any edge $e \in G$, we use $w_G(e)$ to denote its weight. We drop the dependence on $G$ when it is clear from context. For any graph $G$, we use $d_G(.,.)$ to denote the shortest path metric in the graph. If two vertices $u$ and $v$ are not in the same connected component, we set $d_G(u, v) = W'$ where $W' \gg nW$ is a large integer. For any two vertices $u$ and $v$ and a tree $T$ we will use $P_{u,v}^T$ to denote the (unique) path between $u$ and $v$ (assuming both vertices are in the same component), dropping the dependence on $T$ when it is clear from the context.

Given a graph $G'$, we will often use $V(G')$ and $E(G')$ to denote the vertices and edges of the graph respectively. Unless otherwise stated, $V$ and $E$ refer to the vertices and edges of the input graph $G$ which is undergoing edge insertions and edge deletions.

**Embedding.** We use $\rho : V \rightarrow V'$ to denote an embedding from a metric space $(V, d)$ to a metric space $(V, d')$. In this paper, we will generally work with *random* embeddings.

These are embedding functions $\rho$ that are chosen, randomly, from a larger set $P$. Importantly, the function $\rho$ itself is deterministic. i.e., querying $\rho(v)$ multiple times will always lead to the same answer. Our choice of the function $\rho$ will be random.

We say a (randomly chosen) embedding $\rho$ has *expected distortion* $\alpha$ if, for any $u, v \in V$,

$$d(u, v) \leq \mathbb{E}\left[d'(\rho(u), \rho(v))\right] \leq \alpha d(u, v)$$

and say it is *non-contractive* if

$$\forall u, v : d'(\rho(u), \rho(v)) \geq d(u, v).$$

In this paper, we will consider mainly two target metric spaces $V'$ for the output embedding. The first choice metric embedding into trees, or more generally forests. The second choice is the set of real-valued vectors $\mathbb{R}^k$ with the $\ell_p$ norm $\|x - y\|_p := \left(\sum_{i=1}^{k}(x[i] - y[i])^p\right)^{1/p}$ where $x[i]$ denotes the $i$-th coordinate of $x$. We note that distances can be infinite for graphs and forests if two vertices are not in the same connected component. However, for any two points $x, y$ in $\mathbb{R}^k$, we have $\|x - y\|_p < +\infty$. Given this discrepancy, when considering embedding into $\ell_p$ spaces, we define the input metric $d_G(., .)$ as the shortest path metric for any two vertices that are in the same component, and define it as $W'$ for any two vertices that are not in the same connected component of $G$. This is essentially stating that even if there is no edge $e \in E$ between two vertices, one can still move from one to the other with weight $W'$. Note that, since $W' \gg nW$ and edge weights are in range $[W]$, the distance between any two vertices that are in the same component is less than the distance between two different components.

**Dynamic model.** We consider a *fully dynamic* model in which the edges of a graph $G = (V, E)$ are inserted and deleted, and the goal is to always maintain an embedding into the $\ell_p$ space. In this work, we operate in the *oblivious adversary* model for dynamic algorithms which is a standard model for dynamic algorithms (Henzinger et al., 2018; Forster et al., 2021; Banihashem et al., 2024). This means that an adversary chooses the update stream with knowledge of our algorithm, but without knowing the random bits we use or the output embedding we maintain. This is equivalent to assuming that the adversary specifies all of the updates before our algorithm is started.

Our algorithm will maintain an embedding function $\rho$ which can efficiently be queried. This is in contrast to *explicit maintenance* of an embedding where the algorithm is required to explicitly output all changes to the embeddings of the vertices, i.e., which vertices have had their embedding

value changed and what the new value is. In fact, our lower bounds show that, in the model we are considering, explicitly maintaining a low-distortion embedding is not possible. Note that explicit maintenance is possible in a variant of the decremental model where the edge weights increase but there are no deletions (Banihashem et al., 2024).

## 3. Fully Dynamic Embedding

In this section, we present our dynamic algorithm for the problem. Our algorithm assumes access to a fully dynamic algorithm for embedding the metric into trees in which the path between each two vertices contains a *heavy* edge with length comparable to the total path length. Formally, we define the concept of *edge-dominant trees* below.

**Definition 3.1** (edge-dominant trees). We say a tree $T$ is *edge dominant* with parameter $\beta < 1$ if for any path $(v_1, \ldots, v_f)$ in the tree, where $v_i$ are vertices of the tree, there exists an edge $(v_i, v_{i+1})$ in the path satisfying $w_T(v_i, v_{i+1}) \geq \beta d_T(v_1, v_f)$. We say a forest $T$ is edge dominant if all the trees in the forest are edge dominant.

The following lemma shows that an edge dominant forest with low distortion can be maintained dynamically. In order to effectively leverage the existing results for embeddings into trees, for this lemma we assume that the forest maps different connected components of the input graph to separate trees; i.e., we do not assume that there is a "default" edge of weight $W'$ between any two vertices. We will later add these default edges in our algorithm to connect different components.

**Lemma 3.2.** *For any $q \geq 2$, There exists an algorithm which has $m^{1/q+o(1)}$ amortized update time with high probability and embeds the graph into a rooted forest $T$ with the following properties.*

1. *Each tree in the forest has height at most $O(q \log(nW))$.*

2. *For any two vertices $u$ and $v$, we have $d_T(u, v) \geq d_G(u, v)$ and $\mathbb{E}[d_T(u, v)] \leq \alpha_q d_G(u, v)$ where $\alpha_q = O(\log(n))^{2q-1} O(\log(nW))^{q-1}$. Here $d_G(u, v)$ is set to $+\infty$ if $u$ and $v$ are in different components of $G$.*

3. *The forest is edge dominant with parameter $\beta_q = 4 \cdot 3^{-q}$.*

*Throughout its running time, the algorithm outputs all changes made to the forest.*

We refer to Section C for the proof of the lemma which is based on the dynamic embedding of (Forster et al., 2021). While the depth of the tree and the upper bound on expected distances have already been established, our result on the existence of a heavy edge in each path is, to the best of our

knowledge, novel.

To simplify the presentation, we will actually build an embedding that satisfies a scaled version of the guarantees in Theorem 1.1; specifically, we will ensure that $\frac{\beta_q}{2} d_G(u, v) \leq \mathbb{E} [\|\rho(u) - \rho(v)\|_p] \leq \alpha_q \log(n) d_G(u, v)$, where $\alpha_q, \beta_q$ are chosen as in Lemma 3.2. Additionally, we will show that $\|\rho(u) - \rho(v)\|_p \geq \frac{\beta_q}{2} d_G(u, v)$ with high probability. Scaling the embedding by $\beta_q/2$ proves Theorem 1.1.

Given the forest obtained from Lemma 3.2, our algorithm works as follows. We first create a new vertex $s$ and connect the root of each tree in the forest to $s$ with weight $W'$. This effectively turns the forest into a tree, where vertices that used to be in separate components are now in distance $\Theta(W')$. We then multiply each edge $e$ in the tree by a random number $\gamma_e$ chosen uniformly at random from $\{1, 2\}$, obtaining a new tree $T'$.

In order to obtain the embedding $\rho(.)$ of a vertex $v$, we repeat this procedure $k = \Theta(\log n)$ times and set $\rho(v)$ to be the concatenation of the distances of $v$ to the root in each of these trees. As we will see in the analysis, the exact choice of constant behind $\log n$ for choosing $k$ will control the high probability guarantees we provide for non-contractivity; for any $a \geq 1$, we can ensure that the tree is non-contractive with probability $1 - 1/n^a$ by choosing a large enough constant.

For simplicity, we will not store $T'$ directly and make the following changes. Firstly, instead of storing a value $\gamma_e$ for each edge, we store a value $\gamma_v$ for each vertex $v \neq s$ in the tree $T'$. This is clearly equivalent because we can associate each edge in the tree with its lower vertex. Secondly, we store the value of $\gamma_v$ in a hash table, adding and removing values as vertices are created and deleted in $T$. In order to obtain the distance of a vertex from $s$, we simply traverse the path from the node to its root, summing up the weight of the edges multiplied by $\gamma_v$. A formal pseudocode is provided in Algorithm 1.

As we will see in the next section, the multiplication by $\gamma_e$ plays a crucial role in the analysis. In the original tree $T$, it is possible that two vertices have similar distances to the root, even if they are far from each other. This is not acceptable however as we require vertices $u$ and $v$ that are far from each other two have accordingly far values in the embedding. By multiplying $\gamma_e$, we are effectively adding noise to the distances. The size of this noise is roughly equal to $d_T(u, v)$ itself, ensuring that the embedded values are far as well. Furthermore, repeating the procedure $k$ times allows us to ensure that the embedding is non-contractive with high probability.

*Remark* 3.3. While ideally we would want a graph $G$ with

$n$ vertices and no edges to be mapped to a tree $T$ with $n$ vertices and no edges, this is not the case. In order the tree embedding algorithm in Lemma 3.2 to avoid making $\Theta(n)$ operations at initialization due to outputting the "new" vertices $V$ added to the tree, the algorithm does not explicitly create a vertex in the tree for each vertex in the graph at the start of the algorithm. Rather, it only creates a corresponding vertex in the tree once there is an edge connected to the vertex. As such, technically the value $\gamma_v$ may not be available for a vertex $v$ if it is isolated, prohibiting us from calculating $\rho(v)$; specifically, we run into an error in Line 19. This issue can be easily fixed however; if such a value is not available, we sample the value $\gamma_v$ at the query time and add it to the hash table. If a corresponding vertex is created later on, we simply replace the value. Since the operation is $O(1)$, it does not affect the update time bounds.

---

**Algorithm 1:** Dynamic embedding into $\ell_p$ metric.

**1 Function** `Init()`:
    **2**    Set $k = \Theta(\log n)$ ;
    **3**    Initialize dynamic forest embeddings $T_1, \ldots, T_k$ (See Section C) and empty hash tables $H_1, \ldots, H_k$;

**4 Function** `Update(u)`:
    `// Insertion or Deletion`
    **5**    **for** $i \in [k]$ **do**
    **6**      $(V_{\text{add}}, V_{\text{remove}}, ) \leftarrow$ `Update`$(T_i, u)$
         `// Forward to` $T_i$ `and store new and deleted vertices in` $T_i$
    **7**      **for** *each $v$ in $V_{remove}$* **do**
    **8**        Remove $(v, \gamma_v)$ from $H_i$;
    **9**      **for** *each $v$ in $V_{add}$* **do**
    **10**        Sample $\gamma_v$ uniformly at random from $\{1, 2\}$;
    **11**        Add $(v, \gamma_v)$ to $H_i$;

**12 Function** `Query(u)`:
    **13**    **for** $i \in [k]$ **do**
    **14**      $p \leftarrow u, \rho_i(u) \leftarrow 0$;
    **15**      **while** $p \notin \text{root}(T_i)$ **do**
    **16**        Retrieve $\gamma_p$ from $H_i$;
    **17**        $\rho_i(u) \leftarrow \rho_i(u) + w_{T_i}(p, \text{parent}_{T_i}(p)) \cdot \gamma_p$;
    **18**        $p \leftarrow \text{parent}_{T_i}(p)$;
    **19**      $\rho_i(u) \leftarrow \rho_i(u) + W' \cdot \gamma_p$ ;
    **20**    **return** $\rho(u) = (\rho_1(u), \ldots, \rho_k(u))$

---

## 4. Analysis

In this section, we prove our main result, i.e., Theorem 1.1. The update time analysis of the algorithm follows by charging each operation to a corresponding tree operation; since the update time of the trees is bounded as in Lemma 3.2, the

claim follows. Since we maintain $\Theta(\log(n))$ trees, our update times are a factor $\log(n)$ higher than that of Lemma 3.2, but this is absorbed by the $m^{o(1)}$ term in the updates. The only extra operation we make is maintaining $\gamma_e$, which can be charged to the running time of the tree embedding algorithm the changes to $T$ are directly outputted. Since sampling, storing, and removing each $(e, \gamma_e)$ pair is $O(1)$, this only affects the update time by a constant. As for the query time, since the height of tree is at most $O(q \log(nW))$, we can output the embedding in time $O(q \log(nW))$ as well. Note that the extra samplings of $\gamma_v$ specified in Remark 3.3 do not affect this as sampling only takes $O(1)$ time.

We now focus on the analysis of the distortion for our embedding. We will show that for any two vertices $u$ and $v$ and any coordinate $i \in [k]$, the expected difference between $\rho_i(u)$ and $\rho_i(v)$ satisfies the following upper and lower bounds.

$$\frac{\beta_q}{2} d_G(u,v) \leq \mathbb{E}\left[|\rho_i(u) - \rho_i(v)|\right] \leq \alpha_q d_G(u,v). \quad (4)$$

It follows that, for all $p \in [1, \infty]$,

$$\mathbb{E}\left[\|\rho(u) - \rho(v)\|_p\right] \geq \mathbb{E}\left[\|\rho(u) - \rho(v)\|_\infty\right] \geq \frac{\beta_q}{2} d_G(u,v),$$

and

$$\mathbb{E}\left[\|\rho(u) - \rho(v)\|_p\right] \leq \mathbb{E}\left[\|\rho(u) - \rho(v)\|_1\right]$$
$$= \alpha_q \log(n) d_G(u,v).$$

As mentioned in Section 3, rescaling the tree proves Theorem 1.1. We will now separately prove the upper and lower bounds in Equation (4). To keep the notation simple, throughout the proof we will omit the dependence on $i$ and simply write $\rho(v), T, T'$ instead of $\rho_i(v), T_i, T_i'$; this is essentially the same as focusing on the special case $k = 1$, where $i$ is always 1 and $\rho(.) = \rho_1(.)$.

To avoid complicating the argument with corner cases, we assume throughout the proof that $u$ and $v$ are in the same connected component; this allows us to directly leverage the guarantees from Lemma 3.2 without running into the $+\infty$ issue discussed earlier. In Appendix B, we separately handle the case where $u$ and $v$ are in different components.

**Upper bound on expected distance** To establish the upper bound, we note that by the triangle inequality in the tree $T'$, the difference between distances of two vertices from the root is at most their distances in $T'$. We know from Lemma 3.2 however that distances in $T$ (and therefore in $T'$) can be bounded in terms of distances in $G$. As such, $\mathbb{E}\left[|\rho(u) - \rho(v)|\right]$ is can also be bounded in terms of $d_G(u,v)$. Formally,

$$|\rho(u) - \rho(v)| = |d_{T'}(s,u) - d_{T'}(s,v)| \leq d_{T'}(u,v)$$

where we have used, respectively, the definition of $\rho$ and the triangle inequality. Since $\gamma_e \leq 2$, this is at most $2d_T(u,v)$. Therefore, $\mathbb{E}\left[|\rho(u) - \rho(v)|\right] \leq 2\mathbb{E}\left[d_T(u,v)\right]$ is at most

$$O(\log(n))^{2q-1} O(\log(nW))^{q-1} d_G(u,v),$$

where the inequality follows from Lemma 3.2

**Lower bound on expected distance** To establish the lower bound, we begin by observing that since $T'$ is a tree, $\rho(u) - \rho(v)$ can be obtained by breaking the path between $u$ and $v$ into two parts and taking the difference between the length of these parts. Formally, let $P_u$ and $P_v$ denote the paths from the root to $u$ and $v$ respectively and let $P'$ denote the longest sub-path appearing in both $P_u$ and $P_v$. Let $P_u' = P_u \setminus P'$ and $P_v' = P_v \setminus P'$ denote the remainder of the paths for $u$ and $v$.

Observe however that $P_{u,v}' := P_u' + P_v'$ (i.e., the concatenation of $P_u'$ and $P_v'$) is the unique path from $u$ to $v$. Formally, letting $\tilde{u}$ denote the last vertex in $P'$, one can get from the vertex $u$ to $\tilde{u}$ by following $P_u'$ and get from $\tilde{u}$ to $v$ by following $P_v'$. This is indeed a path (i.e., does not contain repeated vertices) because if a vertex, say $\tilde{v}$, appears in both $P_u'$ and $P_v'$, then there is a sub-path in both $P_u'$ and $P_v'$ from $\tilde{u}$ to $\tilde{v}$. Since the tree does not contain any cycles, this sub-path must be the same in both $P_u'$ and $P_v'$, contradicting the assumption that $P'$ is the longest shared sub-path in $P_u$ and $P_v$.

We can therefore rewrite $|\rho(u) - \rho(v)|$ as

$$|w_{T'}(P_u) - w_{T'}(P_v)|$$
$$= |w_{T'}(P') + w_{T'}(P_u') - w_{T'}(P') - w_{T'}(P_v')|$$
$$= |w_{T'}(P_u') - w_{T'}(P_v')|, \quad (5)$$

The first equality holds by the definition of $P_u$ and $P_v$ and the second equality follows from the definition of $P'$, where $w_{T'}(.)$ denotes the weight function in the tree $T'$. Let $e_{u,v}$ denote the edge in $P_{u,v}$ whose weight (in $T$) is at least $\beta_q d_T(u,v)$; such an edge is guaranteed to exist by Lemma 3.2. We will use the noise obtained from multiplying $w(e_{u,v})$ by $\gamma_{e_{u,v}}$ to prove the lower bound. Formally, fix the tree $T$ as well as all values $\gamma_e$ for $e \neq e_{u,v}$. Set $\alpha_e = (-1)^{\mathbb{1}\{e \in P_v\} + \mathbb{1}\{e_{u,v} \in P_u\}}$; i.e., $\alpha_e$ is $-1$ for all $e$ that are in the same part of $P_{u,v}'$ as $e_{u,v}$ and is 1 for all other $e$. Since $P_u' + P_v' = P_{u,v}$, we can rewrite $|w_{T'}(P_u') - w_{T'}(P_v')|$ as

$$|w_{T'}(P_u') - w_{T'}(P_v')| = |w_{T'}(e_{u,v}) - \sum_{e \neq e_{u,v}} \alpha_e w_e|. \quad (6)$$

Since we have fixed the graph $T'$ and $\{\gamma_e\}_{e \neq e_{u,v}}$, the only randomness in the above expression is in $\gamma_{e_{u,v}}$. It is well-known that for a random variable $X$, the value $\alpha$ minimizing

$\mathbb{E}\left[|X - \alpha|\right]$ is the median of $X$ and since $w_T(e_{u,v})$ is fixed, the median of $\gamma_{e_{u,v}}(w_T(u,v))$ is $\frac{3}{2}w_T(u,v)$. Therefore,

$$\mathbb{E}\left[|w_{T'}(P'_u) - w_{T'}(P'_v)|\right]$$

$$= \mathbb{E}\left[\left|\gamma_{e_{u,v}}w_T(e_{u,v}) - \sum_{e \neq e_{u,v}}\alpha_e w_e\right|\right]$$

$$\geq \mathbb{E}\left[\left|\gamma_{e_{u,v}}w_T(e_{u,v}) - \frac{3}{2}w_T(u,v)\right|\right],$$

where the inequality holds by the definition of the median and the second equality holds since $\gamma_{e_{u,v}} \sim \text{Uniform}\{1,2\}$. Using the definition of $e_{u,v}$ and the fact that $T$ is non-contractive, we can further bound this with

$$\frac{1}{2}w_T(e_{u,v}) \geq \beta_q d_T(u,v) \geq \beta_q d_G(u,v).$$

Since the values of $T$ and $\{\gamma_e\}_{e \neq e_{u,v}}$ were assumed to be fixed in the above derivation, we have shown that $\mathbb{E}\left[|\rho(u) - \rho(v)| \mid T, \{\gamma_e\}_{e \neq e_{u,v}}\right] \geq d_G(u,v)$. Taking iterated expectation we conclude that $\mathbb{E}\left[|\rho(u) - \rho(v)|\right] \geq d_G(u,v)$, finishing the proof.

**Non-contractivity.** It remains to show that $\|\rho(u) - \rho(v)\| \geq \beta_q d_G(u,v)$ holds with high probability, where we note that we are once again considering the vectorized embedding $\rho(.) = (\rho_1(.), \ldots, \rho_k(.))$ where $k = \Theta(\log n)$. To prove this, it suffices to show that, for all $i$, with probability at least $1/2$ we have $|\rho_i(u) - \rho_i(v)| \geq \beta_q d_G(u,v)$. Since $k = \Theta(\log n)$, this would imply that with high probability, $|\rho_i(u) - \rho_i(v)| \geq \beta_q d_G(u,v)$ for some $i \in [k]$, finishing the proof.

To prove the above claim, we once again fix the value of $T$ and $\{\gamma_e\}_{e \neq e_{u,v}}$. Since $\gamma_e$ takes the values 1 and 2 with probability $1/2$ each, Equations (5) and (6) imply that $d_T(u,v)$ takes the values $d_1 := |w_T(e_{u,v}) - \sum_{e \neq e_{u,v}}\alpha_e w_e|$ and $d_2 := |2w_T(e_{u,v}) - \sum_{e \neq e_{u,v}}\alpha_e w_e|$ with probability $1/2$ each. We note however that $\max\{d_1, d_2\}$ is at least $\frac{d_1 + d_2}{2}$, and $d_1 + d_2$ is at least

$$\left|w_T(e_{u,v}) - \sum_{e \neq e_{u,v}}\alpha_e w_e\right| + \left|2w_T(e_{u,v}) - \sum_{e \neq e_{u,v}}\alpha_e w_e\right|$$

$$\geq w_T(e_{u,v}),$$

where the second inequality follows from the fact that $|a| + |b| \geq |a - b|$ for all $a, b \in \mathbb{R}$. By definition of $e_{u,v}$, we have $w_T(e_{u,v}) \geq \beta_q d_G(u,v)$, finishing the proof.

## 5. Lower Bound

In this section, we prove Theorem 1.2. We note that the proof also works for the more general setting where $\rho(.)$ maps the vertices to "labels" that are used to calculate distances via some function.

*Proof.* Consider the following "bridge" graph example. We first partition the set of vertices into two parts $V_1$ and $V_2$ with sizes $\lceil n/2 \rceil$ and $\lfloor n/2 \rfloor$ respectively. We then connect all of the vertices that are in the same partition using edges of weight 1. Let $T$ denote the time step at which all the vertices in the same partition are connected. After this step we repeatedly insert and delete a "bridge" edge with weight 1 between the two components. We repeat this step for $T'$ times where $T' \gg T$ is an arbitrary parameter (see Algorithm 2).

Fix any two vertices $u \in V_1$ and $v \in V_2$. Recall that $\alpha$ denotes the expected distortion of the embedding. When the two components are connected, we have $d_G(u,v) \leq 3$. Applying Markov's inequality and using the expected distortion guarantee we obtain

$$\Pr\left[\|\rho(u) - \rho(v)\| \geq W'/2\right] \leq \frac{2\mathbb{E}\left[\|\rho(u) - \rho(v)\|\right]}{W'}$$

$$\leq \frac{6\alpha}{W'},$$

which is at most $1/4$. When the components are not connected, with probability at least $3/4$ the embedding is non-contractive and therefore we have $\|\rho(u) - \rho(v)\| \geq W'$. Taking a union bound, we conclude that for any two consecutive time steps, with probability at least $1/2$, the value of $\|\rho(u) - \rho(v)\|$ needs to change. This in turn means that at least one of the values $\rho(u)$ and $\rho(v)$ needs to change.

For any $t \geq T$, let $\nu_{t,v}$ denote the indicator random variable that takes the value 1 if the embedding $\rho(v)$ changes after time $t$ and let $\nu_t = \sum_v \nu_{t,v}$ denote the number of vertices whose embedding changes. Let $\rho_t(.)$ denote the value of $\rho$ right after the update. We can therefore write $\lceil n/2 \rceil \mathbb{E}\left[\nu_t\right]$ as

$$\lceil n/2 \rceil \sum_{u \in V}\mathbb{E}\left[\nu_{t,v}\right]$$

$$\geq \sum_{u \in V_1}|V_2|\mathbb{E}\left[\nu_{t,v}\right] + \sum_{u \in V_2}|V_1|\mathbb{E}\left[\nu_{t,v}\right]$$

$$= \sum_{u \in V_1}\sum_{v \in V_2}\mathbb{E}\left[\nu_{t,u} + \nu_{t,v}\right]$$

$$\geq \sum_{u \in V_1}\sum_{v \in V_2}\Pr\left[\nu_{t,u} + \nu_{t,v} \geq 1\right]$$

$$= \sum_{u \in V_1}\sum_{v \in V_2}\Pr\left[(\rho_t(u), \rho_t(v)) \neq (\rho_{t-1}(u), \rho_{t-1}(v))\right]$$

$$\geq \sum_{u \in V_1}\sum_{v \in V_2}\Pr\left[\|\rho_t(u) - \rho_t(v)\| \neq \|\rho_{t-1}(u) - \rho_{t-1}(v)\|\right]$$

$$\geq |V_1||V_2| \cdot \frac{1}{2},$$

where the second equality uses the fact that $|V_1|, |V_2| \leq \lceil n/2 \rceil$, the third equality uses the definition of $\nu$ and the final

inequality uses the previously proved claim that the distance $\|\rho(u) - \rho(v)\|$ needs to change with probability at least $1/2$. It follows that, for any $t \geq T$, we have $\mathbb{E}[\nu_t] \geq \Omega(n)$. This in turn implies that the expected number of changes to the embedding in the first $T'$ steps is at least $\Omega(n(T' - T))$. Therefore, the expected amortized update time is at least $\Omega(n(T' - T)/T')$ which is $\Omega(n)$ for $T' \geq 2T$. $\qquad\square$

## 6. Conclusion

We presented the first efficient fully dynamic algorithm for embedding graph metrics into $\ell_p$ spaces, supporting both edge insertions and deletions with efficient update times. Our algorithm has low expected distortion, is non-contractive, and allows efficient retrieval of a queried vertex's embedding. Additionally, we established a lower bound showing that any algorithm explicitly outputting embedding updates cannot simultaneously be non-contractive and achieve low expected distortion, highlighting the necessity of our assumptions.

## Impact Statement

This paper presents work whose goal is to advance the field of Machine Learning. There are many potential societal consequences of our work, none which we feel must be specifically highlighted here.

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

# A. Further Related Work

**Static algorithms.** As mentioned earlier Given the importance of embeddings into $\ell_p$ spaces, many previous works have studied the problem of low-distortion embeddings in the statics setting; we here highlight a few works that are more closely related to our problem. Most notably, (Bourgain, 1985) obtained an algorithm for embedding into an $\ell_p$ space with $O(\log n)$ distortion, which is tight (Linial et al., 1995). For embedding into trees, the seminal work of (Bartal, 1996b) obtained an algorithm with $O(\log^2 n)$ distortion, which was later improved to $O(\log n)$ by (Fakcharoenphol et al., 2003). Additionally, it is possible to obtain an embedding directly from trees into $\ell_2$ with $\sqrt{\log \log n}$ distortion (Linial et al., 1998).

While these works and the techniques they introduce are the main inspiration behind the ideas used in our paper, they do not have any direct implications for our problem because they consider a static setting whereas we study a dynamic setting in which the graph is constantly changing. Rerunning a static algorithm after each update is possible but not efficient; in contrast, we show how to maintain a low-distortion embedding efficiently.

**Dynamic embeddings.** Recent works have studied dynamic embeddings into simple metric spaces. In this line of work, (Forster et al., 2021) obtained a decremental algorithm with $O(m^{1+o(1)})$ update time and $O(\log^2(n) \log(nW))$ expected distortion, as well as a fully dynamic algorithm with $O(m^{1/q+o(1)})$ update time and $O(\log(n))^{2q} O(\log(nW))^{q-1}$ expected distortion. One of the key tools they developed to obtain these results is an efficient decremental algorithm for maintaining Low Diameter Decompositions (LDDs) for a graph. This construction was later used by (Banihashem et al., 2024) who obtained a decremental algorithm for embedding into $\ell_p$ spaces with $O(\log^3 n)$ expected distortion and $O((m^{1+o(1)} \log^2 W + Q \log(n)) \log(nW))$ total update time, where $Q$ refers to the total number of updates. They also showed that efficiently maintaining a low-distortion embedding is not possible if we require that the embedding has low distortion with high probability, and each change to the embedding of a vertex is explicitly outputted by the algorithm. Our work extends these results by obtaining the first fully dynamic algorithm for embedding into $\ell_p$ spaces. Note that the negative result of (Banihashem et al., 2024) does not apply to our setting because **(a)** we do not output changes to the embedding of each vertex and **(b)** our distortion guarantees only hold in expectation. We further strengthen this negative result by showing that even if a fully dynamic algorithm only preserves distances in expectation, as long as the embedding is non-contractive, explicitly outputting changes cannot be done efficiently.

**Dynamic spanning trees.** A closely related line of work considers the problem of maintaining a spanning tree of a graph with low average stretch. For unweighted graphs, was first studied by (Forster & Goranci, 2019), who obtained an algorithm for with $n^{1/2+o(1)}$ update time and $n^{o(1)}$ average stretch. The update time was later improved to $n^{o(1)}$ by (Chechik & Zhang, 2020). In addition, (Chechik & Zhang, 2020) obtained an algorithm for the problem in a decremental setting. A key contribution of this work was a novel pruning technique for obtaining Low Diameter Decompositions (LDDs), which was later leveraged by the aforementioned work of (Forster et al., 2021) to obtain a dynamic embedding into trees.

**Dynamic shortest paths.** Many works have considered dynamic algorithms for maintaining information about distances in the graph in some form. This includes single source shortest paths (Roditty & Zwick, 2004; Henzinger et al., 2016; 2018), distance oracles (Thorup & Zwick, 2005; Abraham et al., 2012; Forster et al., 2023), spanners (Baswana et al., 2012; Bodwin & Krinninger, 2016), and hopsets (Łącki & Nazari, 2020) to name a few. These tools have proved to be instrumental in the development of dynamic embeddings; e.g., the decremental LDD of (Forster et al., 2021) uses the decremental SSSP of (Henzinger et al., 2018). We refer to (Hanauer et al., 2021) for a comprehensive survey. We note that while randomness is crucial for many of the developments in the field, many works have studied deterministic algorithms for these problems as well (Chuzhoy, 2021; Bernstein et al., 2022).

**Online embeddings.** A closely related line of work is *online embeddings* where the elements of a metric arrive sequentially and we need to assign an embedding to them (Indyk et al., 2010; Barta et al., 2020; Newman & Rabinovich, 2023). While the problem statement is similar to the dynamic setting, especially the insertion-only case, the two problems are fundamentally different. For dynamic algorithms, the main focus is on having a low update time and as such the problem is mainly computational. In contrast, the key issue for designing online embeddings, and online algorithms in general, is that the algorithm is obtaining information incrementally, but needs to make decisions irrevocably.

# B. Omitted Proofs

## B.1. Proof of Theorem 1.1

**Distortion across different components.** We now handle the case where $u$ and $v$ are in different components of the graph $G$. Given Lemma 3.2, this means that the two vertices will be in separate trees in each of the forests $T_1, \ldots, T_k$. The upper bound on the distortion follows fairly easily, the difference between the embedding of any two vertices is at most

$$
\begin{aligned}
\|\rho(u) - \rho(v)\| &\leq \sum_{i=1}^{k} |\rho_i(u) - \rho_i(v)| \\
&\leq \sum_{i=1}^{k} |\rho_i(u)| + \sum_{i=1}^{k} |\rho_i(v)| \\
&\leq \Theta(k) \max_{u} \mathbb{E}\left[|\rho_1(u)|\right],
\end{aligned}
$$

where for the final inequality we have used the fact that $T_1, \ldots, T_k$ are all sampled from the same distribution and as such $\rho_i(u)$ are all i.i.d. Given our construction of $T_i$ however, for any $u$, the distance to the root of the tree is at most $O(1)^q nW \leq \alpha_q nW$; this follows from the recursive nature of the tree construction which, in each iteration, can increase the maximum distance by at most a constant factor. This means we can bound the distance of $\rho(u)$ and $\rho(v)$ with $\Theta(\log(n)\alpha_q W')$, which finishes the upper bound proof because $d_G(u,v) = W'$.

As for the lower bounds, we note that since the two same argument as before applies; specifically, if we do not apply the scaling by $\gamma_v$, the path connecting $u$ and $v$ in the tree $T'$ contains two edges that are of weight $W'$. The exact same argument as before implies that the random scaling causes the vertices to be in distance at least $W'$ in both expectation and high probability.

## B.2. Pseudocode for Theorem 1.2

| **Algorithm 2:** Bridge Edge Update Process |
|---|

**Input:** Vertex set $V$ of size $n$, number of bridge toggles $T'$
**Output:** Sequence of graph updates

1 Partition $V$ into $V_1$ and $V_2$ such that $|V_1| = \lceil n/2 \rceil$, $|V_2| = \lfloor n/2 \rfloor$;
2 Add all edges of weight 1 between pairs in $V_1$ and between pairs in $V_2$;
3 Let $T \leftarrow$ time step when intra-partition edges are fully added;
4 Fix any $u \in V_1$ and $v \in V_2$;
5 **for** $t = T + 1$ **to** $T + T'$ **do**
6     **if** *t is odd* **then**
7         Insert bridge edge $(u,v)$ of weight 1;
8     **else**
9         Delete bridge edge $(u,v)$;

# C. Constructing Edge Dominant Trees

In this section, we prove Lemma 3.2. The proof uses the recursive approach of (Forster et al., 2021) for constructing fully dynamic trees. The approach starts with a (slow) fully dynamic algorithm and "bootstraps" it using a decremental algorithm and obtains a speedup. (See also (Abraham et al., 2012; Forster et al., 2023) for similar applications of this technique) If we simply rerun a static algorithm after each update for the fully dynamic algorithm and use the decremental algorithm of (Forster et al., 2021), this leads to an algorithm with update time $m^{1/2+o(1)}$. Repeating this approach with the new dynamic algorithm, we can further improve the running time, at the cost of worse expected distortion. As we show in this section, this fully dynamic algorithm satisfies the edge dominant guarantees required by Lemma 3.2.

The remainder of the section is organized as follows. We first provide the boosting reduction of (Forster et al., 2021) and state the reduction we use from prior work (Section C.1). We then show that the tree obtained from the boosted algorithm satisfies the edge dominant guarantee required in Lemma 3.2, provided that the original algorithms used for boosting satisfied the property (Section C.2). Finally, we state a specific instantiation of the boosting framework using the decremental algorithm of (Forster et al., 2021) and the static algorithm of (Blelloch et al., 2016) (Section C.3).

## C.1. Tree Construction

We now present the fully dynamic reduction of (Forster et al., 2021) for boosting a slow fully dynamic algorithm into a faster one by combining it with a decremental algorithm. Specifically, given a decremental algorithm $A$ and a (slower) fully dynamic algorithm $B$, we obtain a new fully dynamic algorithm $C$ whose update time improves on that of $B$.

**Overview of approach.** We run the algorithm $A$ in phases of length $\ell$, restarting the algorithm at the beginning of each phase. In each time step, let $E$ denote the set of edges together with their weight and $F$ denote the same set at the start of the current phase. We will use $I = E \backslash F$ to denote the set of edges that have been inserted since the beginning of the phase but have not yet been deleted and use $D = F \backslash E$ to denote the set of edges that have been deleted since the beginning of the phase.

Let $U$ denote the set of endpoints of edges in $I$; i.e., $U = \{u : (u, v) \in I \text{ for some } v\}$. Let $T_A$ denote the tree maintained by the decremental algorithm and define $P_u$ to be the path from the root to $u$ in $T_A$. Define $P := \cup_{u \in U} P_u$ and $H := P \cup U$. Let $T_B$ be the graph obtained by maintaining a fully dynamic embedding of $H$ using the algorithm $B$. Consider the graph $(T_A \backslash H) \cup T_B$ obtained by replacing $H$ in $T_A$ with the tree $T_B$, and let $T_C$ be the resultig graph.

**Dynamic maintenance.** We next state a lemma form (Forster et al., 2021) that shows the aforementioned values can be maintained efficiently in a dynamic setting. We begin with some definitions. Let $h_A$ and $s_A$ denote upper bounds on the height of $T_A$ and the expected distortion of $T_A$ respectively. Let $h_B$ and $s_B$ denote the corresponding values for $T_B$. Let $\chi_A$ denote an upper bound on the number of times the path from a fixed vertex to the root can change during a run of algorithm $A$. Let $t_A(m, n)$ and $u_B(m, n)$ denote upper bounds on, respectively, the total time of algorithm $A$ and amortized update time of algorithm $B$ when run on a graph with $n$ vertices and at most $m$ edges.

**Lemma C.1** ((Forster et al., 2021)). *The graph $T_C$ is a non-contractive forest with expected stretch bounded by $s_A s_B$, and has height at most $h_A + h_B$. Additionally, for any integer $\ell \geq 1$, there exists a fully dynamic algorithm that maintains the tree $T_C$ with amortized update time $O(t_A(m, n) \log(n)/\ell)$ such assuming at least $\ell$ updates are performed.*

## C.2. Edge Dominant Property

We now show that the tree $T_C$ is edge dominant. On a very high level, this seems reasonable since static embeddings into trees use the framework of Bartal's embedding which is Hierarchical and weights multiply by two when traveling from a leaf towards the root. This idea more or less extends to decremental dynamic algorithms as well.

For the fully dynamic case however, the proof is more subtle since the output tree is actually obtained by combining two separate trees in the algorithm. To prove the lemma, we will show that the path connecting any two vertices in a tree can be divided into at most three parts such that each part lies entirely in one tree (Lemma C.5). This implies the Lemma's statement by considering the maximum edge in each of these parts.

**Lemma C.2.** *Assume that the trees $T_A$ and $T_B$ are edge dominant with parameter $\alpha$, and assume the algorithms $A$ and $B$ preserve connectivity. Then the tree $T_C$ is edge dominant with scaling $\alpha/3$.*

We say a tree embedding $T$ preserves connectivity for $G$ if for any pair of vertices are connected in $G$ if and only if they are connected in $T$. We next show that the tree $T_C$ preserves connectivity. This can be obtained from the distortion guarantees of the forest; since the forest is non-contractive, any two vertices that are not in the same component of the graph should not be in the same component of the tree. As for vertices that are not in the same component, if they fall in different components with non-zero probability, then their expected distortion would be $+\infty$, contradicting Lemma C.1. We provide a simple proof for completeness however as it provides a nice intuition for the remaining proofs.

**Lemma C.3.** *If $T_A$ and $T_B$ preserve connectivity then so does $T_C$.*

*Proof.* First note that if $u$ and $v$ are in different components in the input, then they must be in different trees in the output since $T_C$ is non-contractive. Now, assume that $u$ and $v$ are in the same connected component in $G$. Consider the path connecting $u$ and $v$ in $G$ and label each vertex with a number indicating the component of that vertex in $T_C$. Since $u$ and $v$ have different labels, there must be an edge in the path such that the label of its endpoints is different.

There are now two cases:

1. The edge is in $F$. In this case, since $T_A$ preserves connectivity, the endpoints must be in the same component of $T_A$. This means that they are connected in $T_C$ as well however. Specifically, to get from one endpoints to another, we can follow the path connecting them in $T_A$ and for any edge that is in $P$ (and therefore is not in $T_C$), we can just go through a "detour" path in $T_B$. Such a detour must exist since $T_B$ preserves connectivity.

2. The edge is not in $F$. In this case, the edge must be in the set $I$ which means it appears in $H$. Since $T_B$ preserves connectivity, the endpoints of the edge are connected in $T_B$, which in turn means that they must be in the same connected component of $T_C$.

In both cases, we have shown that the endpoints of the edge were in the same component of $T_C$, contradicting the choice of the edge. It follows that $u$ and $v$ are in the same component, finishing the proof. $\square$

The following lemma states that the tree $T_A \backslash P$ does not connect different components of $T_B$ together. We note that the converse is not true and $T_B$ can connect different components of $T_A$ together; indeed, if an edge is inserted in $G$ connecting two different components, then the components of $A$ are unchanged (assuming the $\ell$-length phase has not ended), and $T_B$ is necessary for connecting the two components in $T_C$.

**Lemma C.4.** *If $u, v \in V(T_B)$ then $u$ and $v$ are in the same connected component in $T_B$ if and only if they are in the same connected component in $T_C$.*

*Proof.* Assume that $u$ and $v$ are in the same component of $T_B$; then they are clearly in the same connected component in $T_C$ as well since $E(T_B) \subseteq E(T_C)$. Conversely, assume that $u, v \in V(T_B)$ are connected in $T_C$; we will show that they are connected in $T_B$ as well. The key insight we rely upon is that if a vertex $v \in V(T_A)$ appears in $U$, then so does the entirety of its path to the root. Therefore, the graph $U$ cannot put two vertices in the same component of $T_A$ in different components. Since $T_B$ preserves connectivity, this means that $T_B$ cannot do this either.

Formally, let $e_1, \ldots, e_f$ denote the edges of the path connecting them. If all the edges are in $E(T_B)$, then the claim clearly holds. Otherwise, let $e_i$ denote the first edge not in $E(T_B)$. Since the edge is not in $T_B$, it must be in $T_A \backslash P$. Continue along the path until we once again reach a vertex in $T_B$. Let $(u', v')$ denote the vertices in $T_B$ that are connected via this sub-path. Since $u'$ and $v'$ are connected via $T_A$, they are connected in $P$ because $P$ contains the path from both of these endpoints to the root. Since $T_B$ preserves connectivity, this means that these endpoints are in the same component of $T_B$ as well. Therefore, there must be a path connecting them in $T_B$. This is a contradiction as such a path would form a cycle with the path in $T_A \backslash P$, contradicting the fact that $T_C$ is a tree. $\square$

**Lemma C.5.** *Fix two vertices $u, v \in V(T_C)$ and let $e_1, \ldots, e_f$ denote the edges of the path connecting them. There exists indices $0 \leq i \leq j \leq f + 1$ such that $e_1, \ldots, e_i$ and $e_j, \ldots, e_f$ are in $E(T_A \backslash P)$ and $e_{i+1}, \ldots, e_{j-1}$ are in $E(T_B)$.*

*Proof.* The key insight behind the proof is that since $(T_A \backslash P)$ cannot connect two components of $T_B$, any path in $T_C$ cannot enter $T_B$ after leaving it. This means that we can break the path into three parts representing before $T_B$, during $T_B$, and after $T_B$ respectively. Formally, let $V(P)$ denote the set of vertices in $P$. If there $P_{u,v}^C \cap V_P$ is empty, then the path must entirely lie in either $T_A \backslash P$ or in $T_B$. This is because in order to get from any vertex in $T_A \backslash P$ to $T_B$, we need to first go through some vertex in $V(P)$. In this case, the lemma clearly holds. Otherwise, let $v_i, v_j$ denote the first and last vertices for the path in $V(P)$. Since the vertices are connected, they are in the same connected component in $T_C$ which, by the previous lemma, implies they are in the same connected component in $T_B$. Therefore, the path connecting them lies entirely in $T_B$, finishing the proof. (see Figure 1 for an illustration).

$\square$

We can now prove Lemma C.2

*Proof of Lemma C.2.* Fix the vertices $u, v \in V(T_C)$. Let $e_1, \ldots, e_f$ denote the edges of the path connecting them. We need to show that there exists a value $i' \in [f]$ such that

$$w(e_{i'}) \geq \frac{\alpha}{3} d_{T_C}(u, v)$$

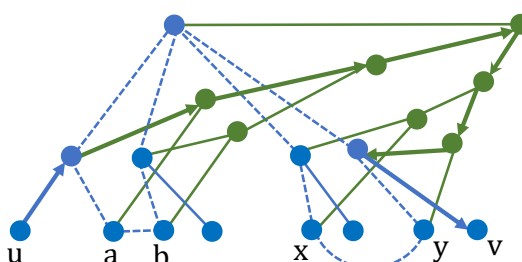

*Figure 1.* Example for Lemma C.5. The blue vertices and edges denote the tree $T_A$. Edges $(a, b)$ and $(x, y)$ have been inserted after the start of the current phase. The dashed edges denote the edges of the graph $U$ and the green vertices and edges denote the tree $T_C$; note that the leaves $T_C$ are in $T_A$. In order to get from $u$ to $v$, we start from the vertex $u$ and following the path to the root until reaching the vertex in $T_B$. Next, we follow the path in $T_B$ to the appropriate leaf and go downwards until reaching $v$.

Let $i$ and $j$ be the indices specified by Lemma C.5. Define the sub-paths $P_{1,i} \in (e_1, \ldots, e_i)$, $P_{i,j} \in (e_i, \ldots, e_j)$, and $P_{j,f} \in (e_1, \ldots, e_i)$. Let $e_{1,i}$, $e_{i,j}$, and $e_{j,f}$ denote the heaviest edges in each sub-path; if any sub-path is empty, set the corresponding edge to an arbitrary value such as `Null` and define its weight to be 0.

Since $T_A$ and $T_B$ are edge dominant with parameter $\alpha$, and each of the three sub-paths is entirely contained in either $T_A$ or $T_B$, we have $w(e_{1,i}) \geq \alpha w(P_{1,i})$ and similarly for $e_{i,j}$ and $e_{j,f}$. Therefore,

$$
\begin{aligned}
\max &\left\{ w(e_{1,i}), w(e_{i,j}), w(e_{j,f}) \right\} \\
&\geq \frac{1}{3} \left( w(e_{1,i}) + w(e_{i,j}) + w(e_{j,f}) \right) \\
&\geq \frac{\alpha}{3} \left( w(P_{1,i}) + w(P_{i,j}) + w(P_{j,f}) \right) \\
&= \alpha \frac{d_{T_C}(u, v)}{3},
\end{aligned}
$$

where the first inequality holds because $\max \geq$ average and the second inequality holds by Edge-dominance of $T_A, T_B$. $\quad\square$

### C.3. Proof of Lemma 3.2

Combining the above results, we obtain Lemma 3.2. The lemma follows using the same proof as Theorem 4.2 in (Forster et al., 2021). We sketch the proof for completeness, emphasizing the edge-dominant property.

*Proof.* We can firstly assume that an upper bound on $m$ because we can start with the value 1 and, each time the upper bound is violated, multiply it by 2 and restart the procedure; since the upper bounds increase exponentially, the restarting cost can always be charged back to the previous updates at the cost of a constant factor.

Assuming an upper bound is known, the proof follows by induction on $i$. The base case follows from (Blelloch et al., 2016) who obtain an efficient algorithm for calculating the FRT tree of (Fakcharoenphol et al., 2003). We note that the tree is edge-dominant with parameter $1/4$. For $q \geq 2$, we use Lemma C.1. Specifically, set $A$ to be the decremental algorithm of (Forster et al., 2021) and set $B$ to be the fully dynamic algorithm obtained by induction for $q - 1$. As long as there are less than $\ell = m^{1-1/q}$ updates, we can simply run the fully dynamic algorithm. Afterwards, we run the recursive algorithm of Section C.1.

The stretch dominant property holds before we reach $\ell$ updates because of the induction hypothesis. As for after the $\ell$ updates, the fully dynamic algorithm satisfies the property with $\beta_{q-1}$. Additionally, the decremental algorithm $A$ satisfies it with parameter $1/4$. Lemma C.2 now implies that the recursive algorithm satisfies the property with parameter $\min \left\{ 1/4, 3^{-(q-1)} \right\} \cdot \frac{1}{3} \geq 3^{-q}$.

$\quad\square$

