# OpenReview forum: "Fully Dynamic Embedding into $\ell_p$ Spaces"
_ICML.cc/2025/Conference — ICML 2025 poster_

### Official Review · Reviewer_srig · 2025-02-15

**Overall Recommendation:** 2

**Summary:**

The Authors presented an algorithm to embed dynamic weighted graph to $\\ell\_p$ space, achieving $O(\\log(n))^{2q} O (\\log (nW))^{q-1}$ expected distortion with $O(m^{1/q + o(1)})$ update time and $O(q \\log (n) \\log (n W))$ query time.

## update after rebuttal
First, I appreciate the authors' sincere responses. However, regrettably, I have decided to keep the original score.
- The authors have failed to answer my crucial concern directly: *why can converting a dynamic graph to $\ell^p$ representations contribute to the ICML community?* The paper [1] is an ICML paper, but this paper again does not explain why they are meaningful in the machine learning community. Moreover, to the best of our knowledge, the paper has not been cited well in the community, which implies that the paper failed to explain its contributions in the ICML community. By the way, the current manuscript cites the arxiv version of [1], which I recommend you modify so that it cites the ICML version, as citing conference or journal versions (if exist) is a convention in the machine learning community.
- The authors try to provide examples [2]-[5]. However, though they are dynamic OR graph embedding, they are not dynamic graph embedding, i.e., not dynamic and graph embedding simultaneously, so they do not explain the importance of the object of the authors' analysis.
- The authors provided experimental results, but they are not designed to answer my question. Although the authors stressed that the distortion was low and the percentage of non-contractive node pairs was also low, they can be zero if we simply use the original graph without embedding. Hence, they do not explain why we need to embed a dynamic graph, unfortunately.

Whether the paper is accepted to this ICML or not, I encourage the authors to clarify before its publication how dynamic graph embedding can contribute to the machine learning community. If you spend time, you may find good application areas where dynamic graph embedding plays a crucial role. Such explanations are necessary to accept theory papers, as analysis on useless things is again useless (One possible exception is where we rigorously prove that something is useless, even though it looks useful.).

[1] Banihashem, Kiarash, MohammadTaghi Hajiaghayi, Dariusz Rafal Kowalski, Jan Olkowski, and Max Springer. Dynamic Metric Embedding into lp Space. ICML 2024

[2] Cohen-Addad, V., Lattanzi, S., Maggiori, A., & Parotsidis, N. (2024). Dynamic correlation clustering in sublinear update time. ICML 2024

[3] Bhattacharya, S., Lattanzi, S., & Parotsidis, N. (2022). Efficient and stable fully dynamic facility location. NeurIPS 2022

[4] Lattanzi, S., Mitrović, S., Norouzi-Fard, A., Tarnawski, J. M., & Zadimoghaddam, M. (2020). Fully dynamic algorithm for constrained submodular optimization. NeurIPS 2020

[5] Cohen-Addad, V., Hjuler, N. O. D., Parotsidis, N., Saulpic, D., & Schwiegelshohn, C. (2019). Fully dynamic consistent facility location. NeurIPS 2019

**Claims And Evidence:**

Despite the interesting and solid theorems, the current manuscript has significant issues, which makes the claims of the paper vague.
- Is considering $\\ell\_p$ included the motivation of the paper, or is it just a tool to achieve a low distortion with small update and query time complexities? The abstract says that "Theoretically, the classic problem in embedding design is mapping arbitrary metrics into $\\ell\_p$ spaces while approximately preserving pairwise distances." The first sentence of the fourth paragraph of the Introduction section says "In this work, we focus on the problem of dynamically embedding into $\\ell\_p$ spaces" without saying like "To solve the ... issues," so I assume that considering $\\ell\_p$ space is also included in the motivation. If so, the paper should have clarified why we are interested in $\\ell\_p$ space. In other words, why are we dissatisfied with just having the graph itself? Or, why do we not consider other spaces, like hyperbolic space? I understand the advantages of using $\\ell\_p$ space from time complexity and low distortion perspectives, but the motivation description in the Introduction section does not say it is the reason why the Authors focus on the $\\ell^p$ space.
- Problem setting is not clearly formulated. Hence, it is hard for readers to judge whether the Authors' claims are sound or not. What oracle is available? Do we know the full vertices and edges and weights initially? How many edges are allowed to be appended at once? We can guess those problem settings by reading the whole paper, but such workloads would not be needed with clear problem-setting descriptions in one place, before mentioning the proposed algorithm.
- One fatal issue of the current manuscript is that it does not clarify how the paper contributes to the ICML community, where machine learning is the main focus, as its name suggests. What do we learn by obtaining the distorted metric, e.g., in $\\ell\_p$ from fully available weighted graph data? Why are we dissatisfied with the graph? In some representation learning settings, like TransE (Bordes et al., 2013) or Poincare embeddings (Nickel & Kiela, 2017), we assume the graph is somewhat noisy or incomplete, but through representation learning, they can correct or complete the original data. This procedure can be called "learning." However, regrettably, from a machine learning viewpoint, the Authors' proposed method **just distorts the original graph without providing any beneficial information**, although the method is still interesting as a *data structure*. I do not dare to call it "machine learning" if it just converts the original data to another form with distortion. Of course, even if it does not directly solve a machine learning problem, the ICML would accept the work that has the potential to contribute to the machine learning community, provided that the paper explains the potential. However, the current manuscript does not clarify the potential. Actually, the proposed "data structure" may have the strong potential to accelerate machine learning on weighted graph data. However, such perspectives are not included in the Introduction section. For the current manuscript to be accepted to the ICML, it would require rewriting from scratch, which is not what the rebuttal period aims at.
- Even as a data structure paper, it needs to be clarified in what application situations we prioritize the update speed and space complexity by sacrificing the accuracy and query time complexity, since 0 distortion and constant query time can be achievable by the (possibly lazy-updated) distance matrix.

Bordes, Antoine, et al. "Translating embeddings for modeling multi-relational data." Advances in neural information processing systems 26 (2013).

Nickel, Maximillian, and Douwe Kiela. "Poincaré embeddings for learning hierarchical representations." Advances in neural information processing systems 30 (2017).

**Essential References Not Discussed:**

Mentioned in other parts.

**Experimental Designs Or Analyses:**

In this paper, Theorems work as verifications of the superiority of the proposed method.
- This is also a presentation issue, but the current version of Theorem 1.1. is too weak and has no practical implications. The existence of something does not imply we can construct or implement it. To claim that we have solved some issues in computer science, including machine learning, we need to state that like "Algorithm 1 satisfies ..." instead of "there exists an Algorithm A that..."
- Having said that, the coexistence of Theorem 1.1 and Theorem 1.2 is a strong point of this paper in that it makes the reason for the choice of Algorithm 1 convincing.

**Methods And Evaluation Criteria:**

In this paper's case, empirical evaluations would be necessary, while no empirical evaluations are included in the paper. The paper is not like a theoretical analysis of existing established methods or problem settings, where empirical evaluations are not effective. Rather, the Authors have introduced a new problem setting, where edges can both increase and decrease. Hence, the authors are responsible for showing some machine learning problem instances that can demonstrate that the new problem setting is worth considering and that the proposed method works.

**Other Comments Or Suggestions:**

- Define $m$ and $n$.
- As discussed above, the current manuscript does not clarify the potential of the paper to contribute to the ICML community. However, as a general computer science (not machine learning but data structure) paper, the theoretical contributions are solid and almost complete. Specifically, providing a data structure and its distortion bound, update complexity, and query complexity are attractive. I would like to humbly suggest the following:
  - If the Authors still want to aim to submit the work to the machine learning community, please rewrite the paper from scratch to clarify its benefit to the community. For example, (Jain et al., 2016) and (Suzuki et al., 2023), published in the machine-learning context, discussed the embedding problem theoretically in the machine-learning context by formulating it as a prediction problem, a typical machine-learning setting. The latter paper even discusses $\\ell\_p$ embedding. Associating the Authors' work with those studies could make your work's contributions in the machine learning context convincing. Note that those papers discuss static cases only, so the Authors' contribution, which discusses dynamic cases, would still be significant.
  - Having said that, let me humbly suggest the possibility of submitting the Authors' work to another venue that can appreciate the Authors' work better. In any case, I encourage you to upload your work to some online repositories to secure your priority rights if you have not.

Jain, Lalit, Kevin G. Jamieson, and Rob Nowak. "Finite sample prediction and recovery bounds for ordinal embedding." Advances in neural information processing systems 29 (2016).

Suzuki, Atsushi, et al. "Tight and fast generalization error bound of graph embedding in metric space." International Conference on Machine Learning. PMLR, 2023.

**Other Strengths And Weaknesses:**

- Theoretical analyses of this paper are interesting and may lead to the proof of upper bound/lower bound in other settings in the field. Especially, the proof of the lower bound, similar to that of the No Free Lunch theorem (in statistical learning theory) seems elegant to me.

**Questions For Authors:**

- Could Theorem 1.2. not be generalised to a (general) metric space? If we can generalize it, it would strongly motivate the choice of $\\ell\_p$ space against all the other metric spaces, including sphere, torus, hyperbolic space, etc.

**Relation To Broader Scientific Literature:**

Nothing to note here.

**Theoretical Claims:**

I could not check the details of the proof since even the problem setting is not formulated in the paper. However, I could not find absurd results as far as I checked.

---

> ### Author Rebuttal · Authors · 2025-03-31
>
> We thank the reviewer for their detailed reading of our paper.
> We are glad that the reviewer found our theoretical analysis "interesting" and "elegant".
> We address the reviewer's concerns and comments below.
>
> > how the paper contributes to the ICML community ... require rewriting ..
>
> We have discussed these motivations in the first 3 paragraphs of the introduction.
> We understand the reviewer's concerns and have revised the paper by incorporating the following text.
> (To keep the paper within the page limit, we have slightly shortened the overview of techniques section and deferred the proof of non-contractivity to the appendix.)
>
> **Embeddings** are a cornerstone of modern machine learning, powering models from **word2vec** to **large language models (LLMs)**. These methods embed discrete data—like words or tokens—into continuous spaces, enabling geometric reasoning about semantics. **Embedding-based architectures** have driven advances in language understanding and generation.
>
> Among embedding targets, $\ell_p$ spaces (especially $\ell_2$) have emerged as the **standard representation space** in modern ML. This is due to (a) their **interpretability**—distances and angles have clear geometric meaning—and (b) the fact that many key primitives are **much more efficient** in $\ell_p$ spaces. For example, **nearest neighbor search** in high dimensions is routinely performed using **locality-sensitive hashing (LSH)**, designed specifically for $\ell_2$ and other vector norms. Similarly, fast kernel approximations, attention mechanisms, and geometric reasoning in LLMs all leverage the vector structure of $\ell_p$ spaces.
>
> Despite their practical dominance, **our theoretical understanding of
> embeddings remains limited**. Data in domains like NLP is **high-dimensional**,
> **structurally complex**, and hard to formalize. Theoretical work addresses
> this via clean abstractions like **graph metrics**.
> Such abstractions play a role similar to that of **PAC learning** in classification theory: while not modeling every
> neural detail, they offer a principled framework for understanding fundamental capabilities and limits.
>
> The **dynamic nature** of data in modern systems further motivates our work. In applications involving **LLMs**, the relevant context or knowledge can shift rapidly. **Embedding methods must therefore adapt efficiently**. Traditional techniques, however, are typically **static** and require expensive recomputation. Our work initiates the study of **fully dynamic embeddings into $\ell_p$ spaces**, a widely used metric class in ML. Modeling data changes via a **dynamic graph** lets us explore embeddings with **provable guarantees** on **distortion**, **non-contractivity**, and **efficiency**. While theoretical, our formulation reflects real challenges in streaming or interactive systems—where representations must update continually without compromising structure or tractability.
>
> > ... empirical evaluations would be necessary ...
>
> We have performed experiments evaluating our embeddings here:  dropmefiles.com/dfsfg
>
> We emphasize again that the main focus of our paper is providing a **theoretical understanding** of dynamic embeddings.
>
> > What oracle is available? ... How many edges are allowed to be appended at once?
>
> We assume that the graph is stored in memory using a standard **adjacency list** representation. **Edge updates (insertions or deletions) occur one at a time**, and the embedding is updated immediately after each modification. That said, our algorithm and analysis are robust enough to conceptually handle **small batches of updates**—for instance, a **polylogarithmic number of changes**—without meaningfully affecting guarantees.
>
> > Current version of Theorem 1.1
>
> While we believe our current phrasing follows standard presentation in the literature, we **agree that explicitly associating Theorem 1.1 with Algorithm 1** would improve clarity.
> The paper has been revised accordingly.
>
> > Could Theorem 1.2 not be generalized ...
>
> We thank the reviewer for this excellent question! Indeed, the lower bound in Theorem 1.2 applies to any class of embeddings based on **labeling**—that is, embeddings where each point is mapped to a label (e.g., a coordinate vector or symbolic descriptor), and distances are computed solely from these labels. For example, points could be mapped to coordinates on a sphere, with distances measured via **spherical distance**.
> We will apply the generalization in our revision.
>
> If you feel our response has adequately addressed your major concerns, we would appreciate it if you possibly adjust your score accordingly.

---

> > ### Comment · Reviewer_srig · 2025-04-05
> >
> > Thank you for your detailed answer. Unfortunately, my main concern remains unresolved.
> > -  word2vec and large language models (LLMs) are embedding from symbol sequences, and associated graphs are not given as an oracle. Those cases might require embedding, but in your problem setting, a graph (adjacency lists) is given. They are totally different cases.
> > - **interpretability, efficient, nearest neighbor search**: Probably, the original graph can do them better than embedding. It does not explain why embedding is important.
> >
> > So, (to put it simply) why can converting a dynamic graph to $\\ell^p$ representations contribute to the ICML community? I could not find such an explanation in the *first 3 paragraphs of the introduction*. The papers you cited are not published in the machine learning community and do not seem to explain why they are beneficial in the machine learning context. They do not seem to deal with dynamic settings, either.
> >
> > > *the main focus of our paper is providing a theoretical understanding of dynamic embeddings.*
> >
> > Focusing on a theory may be accepted when the object (in this case, embedding a dynamic graph to $\\ell_p$ space) of the theoretical analysis is interesting to the ICML community. However,  Hence, you need to cite a paper that discusses embedding a dynamic graph in the machine learning context, or you should demonstrate its benefits through experiments yourself. By the way, the URL (dropmefiles.com/dfsfg) you provided did not work. Could you provide the outline of your experiments?

---

> > > ### Author Response · Authors · 2025-04-06
> > >
> > > Thank you for your follow-up.
> > >
> > > > Focusing on a theory may be accepted when the object ... is interesting to the ICML community.
> > >
> > > We believe this *is* the case. We emphasize that **our work addresses a clear limitation in prior ICML work** [1] and builds on the **fully dynamic model**, which is widely studied in the ML community [2–5]. Dynamic data is central to modern ML, especially in domains like **social networks** and **knowledge graphs**, where the edge structure evolves over time. Moreover, **embedding graph nodes into $\ell_p$ space is a well-established ML technique**. While word2vec operates on sequences, the same principles apply to graphs and are foundational to methods like **node2vec**, which explicitly adapts word2vec to graph structure via random walks and the skip-gram objective [6].
> > > Generally, embeddings with 100 to 400 dimensions are also used to compress data, with applications extending well beyond nearest neighbors, such as serving as input for training other ML models.
> > > These embeddings are widely used for **node classification**, **link prediction**, and **community detection**, as demonstrated in recent ICML and NeurIPS papers [7–10]. Our work provides the **first provable guarantees** for maintaining such embeddings under fully dynamic updates, addressing a timely and practically motivated gap.
> > >
> > >
> > > Regarding the experiments: we have checked the link and confirm that it is working. We evaluate our method on both synthetic and realistic graphs. Graphs G2 and G3 are obtained from G1 (Erdős–Rényi), and G5 and G6 from G4 (power-law cluster), via random edge insertions and deletions
> > > Embeddings are built using our algorithm based on Bartal-style tree embeddings. The **distortion across all graphs remains between 2 and 4**, and the **percentage of non-contractive node pairs** is consistently low: G1: 2.30%, G2: 1.20%, G3: 1.90%, G4: 2.70%, G5: 2.20%, G6: 4.50%.
> > > These results support the effectiveness of our embedding.
> > >
> > > ---
> > >
> > > **References**:
> > >
> > > [1] Banihashem, Kiarash, MohammadTaghi Hajiaghayi, Dariusz Rafal Kowalski, Jan Olkowski, and Max Springer. *Dynamic Metric Embedding into lp Space.* ICML 2024
> > >
> > > [2] Cohen-Addad, V., Lattanzi, S., Maggiori, A., & Parotsidis, N. (2024). *Dynamic correlation clustering in sublinear update time*. ICML 2024
> > >
> > > [3] Bhattacharya, S., Lattanzi, S., & Parotsidis, N. (2022). *Efficient and stable fully dynamic facility location*. NeurIPS 2022
> > >
> > > [4] Lattanzi, S., Mitrović, S., Norouzi-Fard, A., Tarnawski, J. M., & Zadimoghaddam, M. (2020). *Fully dynamic algorithm for constrained submodular optimization*. NeurIPS 2020
> > >
> > > [5] Cohen-Addad, V., Hjuler, N. O. D., Parotsidis, N., Saulpic, D., & Schwiegelshohn, C. (2019). *Fully dynamic consistent facility location*. NeurIPS 2019
> > >
> > > [6] Grover, A., & Leskovec, J. (2016). *node2vec: Scalable Feature Learning for Networks*. KDD 2016
> > >
> > > [7] Davison, A., Morgan, S. C., & Ward, O. G. (2024). *Community Detection Guarantees Using Embeddings Learned by Node2Vec*. NeurIPS 2024
> > >
> > > [8] Abu-El-Haija, S., Perozzi, B., Al-Rfou, R., & Alemi, A. (2018). *Watch Your Step: Learning Node Embeddings via Graph Attention*. NeurIPS 2018
> > >
> > > [9] Zhang, M., & Chen, Y. (2018). *Link Prediction Based on Graph Neural Networks*. NeurIPS 2018
> > >
> > > [10] Baek, J., Lee, D. B., & Hwang, S. J. (2021). *Neo-GNNs: Neighborhood Overlap-aware Graph Neural Networks for Link Prediction*. NeurIPS 2021

---

### Official Review · Reviewer_vp4D · 2025-03-01

**Overall Recommendation:** 3

**Summary:**

This paper studies the problem of maintaining a low-distortion embedding from the shortest path metric on a graph into $\ell_p$ metric, where the graph undergoes edge insertions and deletions. Given a parameter $q$, the paper presents an algorithm that dynamically maintains an embedding that is non-contractive with high probability and admits an expected distortion of $O(\log(n))^{2q} O(\log(nW))^{q - 1}$, where $W$ is the maximum edge weight. Moreover, the algorithm admits an amortized update time of $m^{1/q + o(1)}$ with high probability and only maintains the embedding implicitly with the time of querying the embedding of each vertex equal to $O(q \log (nW) \log n)$. On the other hand, this paper establishes the corresponding negative result, showing that any algorithm that achieves non-contractivity with a constant probability and a sublinear expected distortion and maintains the embedding explicitly must have $\Omega(n)$ amortized update time.

**Claims And Evidence:**

The claims are all supoprted with proofs.

**Essential References Not Discussed:**

No.

**Experimental Designs Or Analyses:**

There is no experiment in the paper.

**Methods And Evaluation Criteria:**

The methods make sense.

**Other Comments Or Suggestions:**

Minors:
- It's not stated in the introduction that the original metric is induced by the shortest path metric on the graph.
- In Line 350, the definition of $\alpha_e$ should instead be $-\alpha_e$.
- In Line 370-372, the second equality is missing.
- In Line 436, should $\leq W' / 2$ be $> W'$?

Typos:
- Line 22: "problem problem"
- Line 115: $d_G(u, v) / 2$ -> $d_G(u, v)$
- Line 138: $s_i$ -> $S_i$.
- Line 184: "note" -> "not"
- Line 198: "as" -> "a"
- Line 176: "$u$ and $v$ not in"
- Line 267: $\beta^{-1}$ -> $\beta$
- Line 291: $E_{add}$ -> $V_{add}$
- Line 343: ". where"
- Line 359: $\gamma_{e, v}$ -> $\gamma_{e_{u, v}}$
- Line 367: $w_T(u, v)$ -> $w_T(e_{u, v})$
- Line 435: $7/8$ -> $3/4$

**Other Strengths And Weaknesses:**

This paper is well-structured and the expositions are clear, although it deserves another round of proofreading. Also, the paper defers the discussion of most relevant literature to the appendix, which is an inappropriate use of the appendix.

This paper generalizes the result in prior work to the fully dynamic setting, and the ideas of maintaining edge-dominant trees and randomly perturbing the edge weights are interesting. However, the negative result is less interesting as the proof inherits the instance from prior work and the analysis is quite straightforward. Overall, the technical contribution of this paper is limited.

**Questions For Authors:**

I don't have further questions.

**Relation To Broader Scientific Literature:**

This paper gives the first algorithm that maintains a low-distortion embedding from a graph to $\ell_p$ metric in the fully dynamic setting, whereas the algorithms in prior work are only for the decremental setting. This paper also extends the lower bound in prior work, which holds for algorithms that have high-probability distortion guarantees, to hold for algorithms that only have expected distortion guarantees. Moreover, the proof of this paper relies heavily on the dynamic tree embedding of prior work.

**Theoretical Claims:**

The proofs are correct to the extent that I have checked (all the proofs in the main body).

---

> ### Author Rebuttal · Authors · 2025-03-31
>
> We thank the reviewer for their careful reading of our paper.
> We are glad that the reviewer believes our "expositions are clear" and
> that "the ideas of maintaining edge-dominant trees and randomly perturbing the edge weights are interesting".
> We address the reviewer's concerns and comments below.
>
> > Discussion of most relevant literature ...
>
> We thank the reviewer for this suggestion. In the current submission, our goal was to introduce the most relevant prior work directly in the **Introduction** as part of the motivation, and then focus the main body of the paper on our technical contributions. Given space constraints, we chose to defer the more exhaustive **Related Work** section to the appendix, as is common in theory papers. That said, we understand the reviewer’s concerns and have revised the paper to ensure that key citations now appear in the main text. Specifically, we now include a concise summary of relevant prior work in the **fourth paragraph of the Introduction**, giving a briefer description of low-stretch spanning trees (which are less central to our contributions), and adding a short discussion of **online embeddings**, which are thematically related.
>
> > Proofreading and typos.
>
> We thank the reviewer for bringing this to our attention. We have corrected the noted issues in the revised version and will continue to carefully proofread the paper to catch any remaining small errors.
>
> > Negative result is less interesting.
>
> We respectfully disagree with this assessment. As we explain in the paper, the **negative result serves an important role**: it demonstrates the **tightness of the assumptions** underlying our positive result. While the lower bound builds on prior ideas, our version extends the result to a more general setting, in particular to **embeddings with low expected distortion that are non-contractive with high probability**. This generalization is not only technically meaningful, but also conceptually clarifying—it helps justify the constraints and guarantees we adopt in our main algorithmic result.
>
> > ... relies heavily on prior work.
>
> While our construction draws on prior work on **tree embeddings**, we do not view this reliance as a weakness. On the contrary:
> - Our analysis of **vector embeddings** uncovers **nontrivial properties** of the tree embeddings used in dynamic settings—properties that, to the best of our knowledge, have not been explicitly documented before. Given the technical depth of the underlying work, these observations are subtle and require careful reasoning.
> - In addition, our paper **establishes a conceptual and technical connection** between dynamic tree embeddings and **vector embeddings into ℓₚ spaces**, which had not previously been explored in this context. While such a connection may seem plausible in hindsight, it is far from obvious, especially given the nontrivial obstacles highlighted in the "Low-distortion trees" paragraph of Section 1.2.
> - Finally, we note that prior constructions—such as low-stretch spanning trees—are themselves composed of incremental innovations on existing techniques. In this spirit, our work **adds new insights** and **extends the applicability** of known ideas to a new and natural problem setting.
>
>
> If you feel that our response has adequately addressed your major concerns, we would appreciate it if you possibly adjust your score accordingly.

---

> > ### Comment · Reviewer_vp4D · 2025-04-02
> >
> > I greatly appreciate your further elaboration on the significance of the results. After a second thought, I decided to raise my score.

---

> > > ### Author Response · Authors · 2025-04-02
> > >
> > > Thank you for raising your score!

---

### Official Review · Reviewer_RMLF · 2025-03-12

**Overall Recommendation:** 3

**Summary:**

The paper presents a fully dynamic algorithm for embedding graph metrics into ℓp spaces, supporting edge insertions and deletions. The algorithm achieves low expected distortion, non-contractivity, and efficient query and update times. Key results include maintaining low-distortion embeddings with O(log(n)) expected distortion and O(m1/q+o(1)) update time. It also demonstrates the impossibility of achieving such properties with explicit update outputs.

**Claims And Evidence:**

The claims made in the paper are generally supported by clear evidence. The theoretical results and algorithmic steps are well-explained, and the authors provide proofs to support their findings. The primary challenge of maintaining low-distortion embeddings in a dynamic setting is tackled effectively with new techniques. However, the distinction between the static and dynamic settings could be elaborated more clearly for readers unfamiliar with this field.

**Essential References Not Discussed:**

There are no major gaps in the references, but a more thorough discussion on the limitations of dynamic embeddings, particularly in the context of real-world applications, would improve the paper's relevance. The exploration of other dynamic graph problems, such as dynamic shortest path problems, could also add depth.

**Experimental Designs Or Analyses:**

The paper does not provide empirical results, which is a limitation. Although the theoretical analysis is thorough, real-world validation of the algorithm's performance in dynamic graphs would strengthen the claims. Including some practical experiments would help in assessing the feasibility of the approach in real applications.

**Methods And Evaluation Criteria:**

The methods are appropriate for the problem at hand.  The algorithm efficiently handles dynamic edge insertions and deletions, which is a challenge in metric embeddings.  The evaluation criteria, such as expected distortion and update time, are well-defined and relevant to the problem.  However, the use of ℓp spaces should be further evaluated by specifical real-world task.

**Other Comments Or Suggestions:**

Including some empirical results or case studies showing the algorithm's performance on real-world dynamic graphs would improve the paper’s impact and reliability.

**Other Strengths And Weaknesses:**

The primary strength of the paper is its novel contribution to dynamic metric embedding, specifically for ℓp spaces. The theoretical guarantees of low-distortion and non-contractivity are significant contributions to the field. The clarity and precision of the theoretical analysis are well. However, the lack of experimental validation and real-world application examples reduces the impact of the work.

**Questions For Authors:**

see above.

**Relation To Broader Scientific Literature:**

The paper positions itself within the existing body of work on metric embeddings, dynamic graph algorithms, and low-distortion embeddings. It builds on the works of Bourgain, Bartal, and Forster, among others.

**Theoretical Claims:**

The paper presents theoretical foundation for the problem. The proofs of expected distortion, non-contractivity, and update/query times are argued. There are no apparent flaws in the theoretical claims, and the methodology for bounding the distortion and ensuring non-contractivity is sound.

---

> ### Author Rebuttal · Authors · 2025-03-31
>
> We thank the reviewer for their thoughtful and constructive review. We appreciate the positive assessment that our results
> are "significant contributions to the field" and that "The clarity and precision of the theoretical analysis are well".
>
> Below, we respond to the reviewer’s specific concerns:
>
> > ... the distinction between the static and dynamic settings ...
>
> Thank you for the suggestion. We have revised **paragraph 4** of the Introduction to further clarify this.
> In short, **static settings** assume the input data is fixed, whereas in the **dynamic setting**, the data changes over time via updates.
> In our model, these updates take the form of insertions and deletions of edges in the underlying graph, which is the standard
> model used in dynamic graph theory.
> The goal is to maintain a good output—here, a low-distortion embedding—even as the input graph evolves.
>
> > The use of $\ell_p$ spaces ...
>
> In practice, $\ell_2$ embeddings are a standard representation in many ML tasks—including **word2vec**, **LLMs**, **kernel-based models**, and **LSH-based nearest-neighbor search**—due to their interpretability and computational efficiency. These embeddings underpin similarity search, clustering, and classification across numerous applications. By focusing on $\ell_p$ spaces, we aim to provide theoretical tools directly aligned with this widely used embedding paradigm.
>
> > ... empirical results ...
>
> We have performed experiments evaluating our embeddings here: dropmefiles.com/dfsfg
>
> We emphasize however that the main focus of our paper is providing a **theoretical understanding** of dynamic embeddings, akin to the role that **PAC learning theory** plays in understanding classification.
> Just as PAC models offer deep insights into learning even when they abstract away the full complexity of modern neural networks, our model—though idealized—helps clarify the possibilities and limitations of embedding dynamic structures.
>
> > ... including pseudocode
>
> We have aimed to provide pseudocode or detailed algorithmic descriptions for all major components in the main body—particularly for **Algorithm 1**. Additionally, we have now included **pseudocode in the appendix for the lower bound construction used in the proof of Theorem 1.2**, to further aid readability.
>
> > ... exploration of other dynamic graph problems, such as dynamic shortest path problems ...
>
> We thank the reviewer for this suggestion and have expanded on this in Appendix A.
>
> If you feel that our response has adequately addressed your major concerns, we would appreciate it if you possibly adjust your score accordingly.

---

### Official Review · Reviewer_kd5G · 2025-03-16

**Overall Recommendation:** 3

**Summary:**

This paper is about dynamic maintenance of Bourgain embeddings (a.k.a. Embedding metric spaces into low-dimensional l_p spaces) with low distortion for undirected graphs with polynomial bounded lengths that undergo edge insertions and deletions. The main result is dynamic Bourgain embedding for graphs that achieve expected stretch O(log n)^{2q} O(log (nW))^{q-1} with O(m^{1/q+o(1)}) update time and O(q \log n \log nW) query time, where is a positive integer larger than 2.

Note here that the embedding of each vertex is maintained only *implicitly*; if one would like to report all the necessary changes to the embedding after each edge update, then there are simple, strong lower-bound that show that even achieving sub-linear expected stretch is out of the question. Just think of a dumbbell graph consisting of two cliques sitting on ~ n/2 vertices, connected by a few edges. One can that simply insert/delete these edges alternatively, which would result in expensive changes in the underlying embedding.

Previous works in the literature could only maintain *explicit* embeddings in the decremental setting, where only length increases are allowed.

The technical contribution of the paper can be viewed as reducing the dynamic Bourgain embeddings to the dynamic tree embedding work of Forster et al. SODA’21 with few additional technical observations: (1) in the trees constructed on these works, the path between any two nodes contains an edge that is *heavy* compared to the shortest path between u and v; (ii) adding some *noise* to the distance estimates, which help to resolve some technical issues about proving that that the embedding is non-contractive.

## update after rebuttal

I appreciate the authors' effort in compiling a very detailed rebuttal. As my score indicates, I'm generally positive about the paper and the paper could be accepted. However, despite the importance of the problem and the strong theoretical guarantees, it is still unclear to me (and to some other reviewers) how dynamic tree embeddings fit within the ICML literature.

**Claims And Evidence:**

The paper is well written. Also, the claims are supported by convincing evidence.

**Essential References Not Discussed:**

I didn't find any.

**Experimental Designs Or Analyses:**

No experiments so nothing to comment here.

**Methods And Evaluation Criteria:**

This is a theory paper, so there is nothing to discuss about the evaluation criteria.

The dynamic model studied in this paper is the standard one in the literature.

**Other Comments Or Suggestions:**

Well written paper, and didn't have a hard time to follow.

Section *Embedding* in Page 4, I think you have messed up *rho* with *d'* -- please double check.

**Other Strengths And Weaknesses:**

Strengths: a simple reduction for dynamic Bourgain embedding with some technical additions which seem to require care. I haven't thought a lot about the problem myself to judge the technical novelty here, but I'd like to emphasize that the paper also uses a quite strong hammer. From that perspective, the paper is not as easy as it may seem. I like the simplicity of the reduction.
The problem is of fundamental important to many communities.

Weaknesses: Maybe discussing the applications where implicit embeddings would make sense?

I lean towards acceptance.

**Questions For Authors:**

None.

**Relation To Broader Scientific Literature:**

Metric embeddings are at the core of many communities without computer science, including image retrieval, computer vision, theory and more recently, machine learning. Dynamic algorithms can be viewed as an effort to design algorithms that are closer to real-world data. Questions at the intersection of both of these topics should be relevant to many communities.

**Theoretical Claims:**

I think all proofs look reasonable to me. I haven't done thorough checks, but at at several places I stopped and tried to follow the math, and it seems correct to me. The overall idea is also sound.

---

> ### Author Rebuttal · Authors · 2025-03-31
>
> We thank the reviewer for their thoughtful and detailed review. We are particularly grateful for the positive assessment that the paper is "well written" and that the problem is "of fundamental importance to many communities." We're also glad that the reviewer appreciated the elegance and care in our technical approach, particularly the reduction to dynamic tree embeddings.
>
> Below we respond to the specific points raised:
>
> > Maybe discussing the applications where implicit embeddings would make sense?
>
> Thank you for this helpful suggestion.
> In many practical settings, maintaining **explicit embeddings** is either infeasible or
> unnecessary. For example:
> - In large-scale systems, memory constraints often prevent storing the entire
>   embedding, and only **pointwise access** to node embeddings is needed at
>   query time.
> - Many downstream tasks—like nearest neighbor search, link prediction, or
>   routing—only require **on-demand access** to distances or individual
>   coordinates, not the full embedding at once.
>
> In such scenarios, **implicit embeddings** provide a flexible and scalable alternative, and our work offers strong theoretical guarantees for maintaining them efficiently in dynamic settings.
>
> > Section Embedding in Page 4, I think you have messed up rho with d' — please double check.
>
> Thank you for catching this. You are correct; the equations should say $d'(\rho(u), \rho(v))$ instead as
> we are calculating the distance of the embedded points in the embedding space.
> We have revised the paper accordingly.
>
> If you feel that our response has adequately addressed your major concerns, we would appreciate it if you possibly adjust your score accordingly.

---

### Decision · Program_Chairs · 2025-05-01

**Decision:**

Accept (poster)

**Comment:**

This paper studies the problem of maintaining an implicit embedding of a graph metric into $$\ell_p$$, under edge insertions and deletions into the graph. The authors achieve this by exploiting work on dynamic tree based embeddings.

The reviewers agree that the results and techniques are interesting, and that this paper makes a valuable contribution to the literature on metric embeddings and dynamic graph algorithms. The authors, however, did not make a convincing case for the relevance of the results to ICML - while embeddings are important to ML, it is not clear that this specific problem of maintaining an embedding of a graph metric under insertions and deletions has any connection to ML.